# Simpler neural networks prefer subregular languages

**Charles Torres**
University of California, Irvine
Language Science
charlt4@uci.edu

**Richard Futrell**
University of California, Irvine
Language Science
rfutrell@uci.edu

## Abstract

We apply a continuous relaxation of $L_0$ regularization (Louizos et al., 2018), which induces sparsity, to study the inductive biases of LSTMs. In particular, we are interested in the patterns of formal languages which are readily learned and expressed by LSTMs. Across a wide range of tests we find sparse LSTMs prefer subregular languages over regular languages and the strength of this preference increases as we increase the pressure for sparsity. Furthermore LSTMs which are trained on subregular languages have fewer non-zero parameters, and LSTMs trained on human-attested subregular patterns have fewer non-zero parameters than those trained on unattested patterns. We conjecture that this subregular bias in LSTMs is related to the cognitive bias for subregular language observed in human phonology, in that both are downstream of a bias for simplicity in a suitable computational description language.

## 1 Introduction

Despite the wide application of neural networks for sequence modeling, their inductive biases remain incompletely understood: for example, whether RNNs have an inductive bias favoring the kind of hierarchical structures used in sentence formation in natural language (Weiss et al., 2018; Ravfogel et al., 2019; Hewitt et al., 2020; McCoy et al., 2020; Edelman et al., 2022). It is important to study the inductive biases of neural networks not only from a scientific perspective, but also a practical one: if neural networks can be developed that have inductive biases toward the same kinds of formal language structures that characterize human language, then these neural networks should be able to learn human language more easily.

Many patterns in natural language are characterized by a particular subset of the regular languages, called **subregular languages**. These languages have been hypothesized to form a bound on learnable human phonological patterns (Heinz, 2018),

as well as patterns within tree structures characterizing syntax (Graf, 2022). The particular subregular languages theorized to be human learnable are strictly piecewise (SP), strictly local (SL), and multiple tier-based strictly local (MTSL). Artificial language learning tasks have shown these languages to be easier to acquire for humans (Lai, 2015; Avcu and Hestvik, 2020; McMullin and Hansson, 2019).

Here we take up the question of inductive biases for subregular languages in LSTMs (Hochreiter and Schmidhuber, 1997), using a relaxation of $L_0$ regularization to study networks that are constrained to be simple in the sense of network sparsity. We train these networks on several sets of toy languages which contrast subregular languages against regular languages in a controlled manner. This technique not only allows us to study the inductive biases of sparse networks, but also to address an outstanding question in the literature on subregular languages: whether there exists a computational description language under which subregular languages are simpler than other regular languages (Heinz and Idsardi, 2013).

To preview our results, we find that simple LSTMs favor subregular languages: such languages can be represented by LSTMs with fewer parameters, and sparse LSTMs favor subregular languages in generalization.

The remainder of the paper is structured as follows: Section 2 gives relevant background on subregular languages, Section 3 gives our general methods for training sparse neural networks under $L_0$ regularization, Section 4 presents our experiments on matched pairs of regular and subregular languages, and Section 5 concludes.

## 2 Background

It is known that phonological constraints should be regular given the most common formalisms in linguistics (Kaplan and Kay, 1994). However, the regular language class is too permissive: many pat-

| Language type | Example language | In language | Not in language |
|---|---|---|---|
| SL | $G = \{ac\}$ | $bc, adc, cad, ...$ | $ac, bac, acd, ...$ |
| SP | $G = \{ac\}$ | $bc, adb, cad, ...$ | $ac, adc, abc, ...$ |
| TSL | $G = \{ac\}, T = \{a, b, c\}$ | $bc, cad, adbc, ...$ | $ac, adc, addc, ...$ |

Table 1: Example subregular languages on the alphabet $\Sigma = a, b, c, d$. Both SP and SL languages are defined by their alphabet and set of forbidden substrings or subsequences ($G$) in their definition. The SL language above forbids the substring $ac$. The SP language above forbids the subsequence $a...c$, in other words, $c$ cannot follow $a$ in a string. TSL languages require a second set ($T$) specifying which characters are preserved in the tier projection. A tier projection function $f_T$ maps strings to strings, preserving only characters in the set $T$. For example, if $T = a, b, c$ then $f_T(adca) = aca$. The restrictions $G$ in a TSL languages are forbidden substrings in the projected strings.

terns not attested in natural language are members of it. The subregular hypothesis is an attempt to improve this bound, lowering it to certain subclasses of the regular languages (Heinz, 2018). There are two versions of the hypothesis—the weak version and the strong version—hypothesizing different subregular languages as the boundary for human learnable patterns. The strong subregular hypothesis asserts that phonological constraints are within the strictly local (SL) and strictly piecewise (SP) subclasses, whereas the weak hypothesis raises this bound to the tier-based strictly local (TSL) languages (the union of multiple constraints being multiple tier-based strictly local or MTSL). These languages are not the only subregular languages, but they are the most important subregular languages for the hypothesis.

What the SL, SP, and TSL languages all share in common is that they are defined by prohibiting features in the strings of that language. The SL and SP languages (corresponding to the strong subregular hypothesis) are defined by sets of prohibited substrings and subsequences respectively. These prohibitions are meant to capture phonological constraints associated with local restrictions (SL) or harmony-like patterns (SP). The TSL languages are slightly more complex than the other two, forbidding substrings after characters have been removed from a string (the retained characters are defined by a second set, usually labeled $T$ for tier). To demonstrate these prohibitions in action examples of these languages are given in Table 1, however curious readers can look to Heinz (2018), Rogers and Pullum (2011), and Heinz et al. (2011) for a rigorous treatment of these languages as well as definitions for other subregular languages which are not part of the subregular hypothesis.

The recent interest in subregular languages and their ability to capture phonological patterns has stimulated some work on the learnability of subregular patterns by RNNs. Results in this area have been mixed. LSTMs and RNNs both show difficulty when trained to predict membership in subregular languages (Avcu et al., 2017). Nevertheless, other work has still shown that the learning of subregular patterns is easier for sequence to sequence RNNs than the learning of other patterns, echoing human-like preferences (Prickett, 2021). Work in this area is still limited.

What can explain these preferences in LSTMs or humans? Previous work has attempted to tie a preference for simplicity to biases in natural language, for example by using the **minimum description length** (MDL) principle, a formulation of Occam's razor favoring simple hypotheses that account for given data, where a hypothesis is simple if it can be described using a small number of bits in some description language (Grünwald, 2000). Can the subregular preference be similarly explained? Heinz and Idsardi (2013) levelled a challenge against accounts of human language learning based on this method in explaining this preference: It is easy to find examples where a subregular and regular language both have the same description length in some computational formalism (for example a finite-state automaton, FSA). While they left open the possibility that a computational formalism may be found where description length can distinguish these examples, they expressed doubt that such a thing could be found. A similar challenge was raised in Rawski et al. (2017) where MDL accounts were challenged to explain learning biases.

Is there a computational representation that, when given access to data consistent with a subregular and regular language, would learn only the subregular pattern by using an MDL method? We attempt to address these challenges. We will show that when controlling for description length (num-

ber of nonzero parameters) in our case studies, neural networks prefer subregular patterns over regular ones on the basis of simplicity, where FSAs do not. That this is true in LSTMs leaves open the possibility that MDL accounts are capable of explaining this bias in other systems, as in human language learning.

## 3 General Methods

A crucial part of our method requires finding optimal tradeoffs between accuracy and network complexity. We operationalize the complexity of a neural network as its number of nonzero parameters, reflecting the size of the subnetwork required to achieve a given accuracy. As such, we would like to use the $L_0$ norm, which counts the number of nonzero entries in the parameter vector $\theta$:

$$\|\theta\|_0 = \sum_{j=1}^{|\theta|} \mathbf{1}_{\theta_j \neq 0}. \quad (1)$$

Ideally, the $L_0$ norm would form a part of our objective function like so:

$$R(\theta) = \frac{1}{N} \sum_{i=1}^{N} L(p_\theta(x_i), y_i) + \lambda \|\theta\|_0, \quad (2)$$

where $L(p_\theta(x_i), y_i)$ is the log likelihood of the $i$'th label $y_i$ given input $x_i$ in the training data under the model with parameters $\theta$. However, the $L_0$ norm is not differentiable, and cannot serve as part of our objective function during training.

We get around this limitation using a relaxation of $L_0$ regularization developed by Louizos et al. (2018). For the $j$'th parameter in our model $\theta_j$, we assign a masking parameter $\pi_j > 0$. This parameterizes a distribution with cumulative distribution function $Q(\cdot|\pi_j)$, from which we can sample a mask value $z_j \in [0, 1]$, which critically can take the values of true 0 or 1. The vector of values $z$ generated in this way is then used as a mask to filter the parameters $\theta$, generating a vector of masked parameters $\tilde{\theta}$ where $\tilde{\theta}_j = z_j \theta_j$.

This parameterization lends us a new objective function where the labels $y_i$ are predicted from the inputs $x_i$ using the masked parameters $\tilde{\theta}$, and the model is penalized for having too few masked

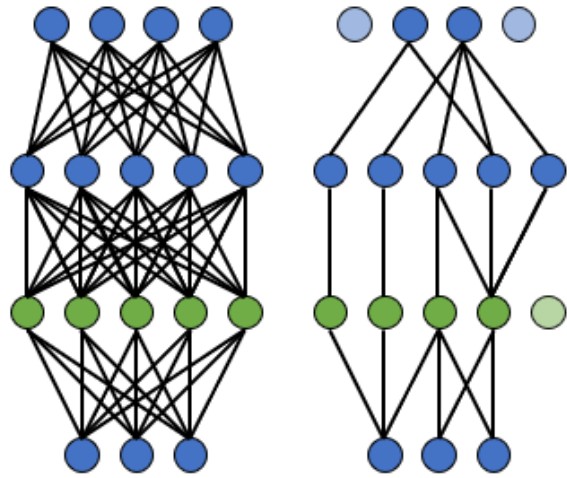

Figure 1: A simple depiction of $L_0$ regularization, LSTM neurons in green,. *Right:* an unregularized network with a similar architecture as in the harmony test *Left:* after $L_0$ regularization has pruned connections.

parameters:

$$R(\theta, \pi) = \mathop{\mathbb{E}}_{z \sim Q(\cdot|\pi)} \left[ \frac{1}{N} \sum_{i=1}^{N} L(p_{\tilde{\theta}}(x_i), y_i) \right] + \quad (3)$$
$$\lambda \sum_{j=1}^{|\theta|} Q(z_j > 0 | \pi_j),$$

where the expectation is over mask vectors $z$ drawn conditional on the vector of masking parameters $\pi$. The regularization term now reflects the expected number of parameters whose value is not masked.

We use Equation 3 in all our experiments below, with different loss functions $L$ as required. When $L$ is a form of cross-entropy this term represents the cost of encoding the data using the distribution specified by the neural network. Under these conditions, the objective becomes a form of minimum description length as a two part code: with the regularization term encoding the distribution, and $L$ encoding the data (Grünwald, 2000). Though our approach focuses on the MDL interpretation, this formulation also has a Bayesian interpretation, optimizing the negative log likelihood with the regularization term as a prior and the loss term as a posterior as we show in Appendix A.

## 4 Experiments

### 4.1 Subregular Battery

The first experiment measures the complexity of LSTMs that learn subregular and regular languages. We call this the **subregular battery**: an experiment

run on three sets of three formal languages, where the description length of the generating FSAs are controlled within each set. Each set of languages contains one strictly local and one strictly piecewise (which are both subregular) and one regular language. One set of languages was used in Heinz and Idsardi (2013)'s argument that minimum description length fails to separate subregular languages from other regular ones; we offer two additional sets. The languages are described in Table 2.

### 4.1.1 Methods

The method can be split into 3 steps: data generation, network training, and evaluation.

**Data generation** For each language, we sample 2000 words from a probabilistic finite state automaton (pFSA) designed to generate words from that language following a geometric distribution on word length. The geometric distribution on word length is important for two reasons: (1) it implies a constant stopping probability in all states, meaning that the distribution over string lengths does not depend on the structure of the strings in a language, and (2) it is memoryless, meaning that the network need not keep track of the total characters observed so far to attain high accuracy. Maintaining a geometric distribution on string lengths is the simplest method for attaining the same distribution over string lengths across languages while sampling from pFSAs.

This data is then split into training, development, and testing sets at an 80/10/10 ratio. Sets are constructed such that no duplicate words existed between sets. This was done by grouping words by form, and then randomly assigning them to the training, testing, or development sets before reshuffling them within the group.

**Network training** Each LSTM has the same architecture: a 3-dimensional embedding layer, a 5 unit LSTM layer, and a 2-layer perceptron decoder with a hidden width of 5 units. We train LSTMs for up to 400 epochs on the training set. For the subregular battery, we use cross entropy loss with the aforementioned $L_0$ regularization technique as our objective function. After each epoch, we evaluate development set performance (including the approximate $L_0$ penalty), and if improvement on it has not increased for 50 epochs, training is halted.[1] Otherwise we use the best performing network on

---

[1]This high patience is necessary due to the stochastic nature of the $L_0$ regularization technique.

the development set over all 400 epochs. We repeat this procedure five times for various regularization penalties to get data on performance at different levels of network complexity.

**Evaluation** Once data is collected, we evaluate the trained networks in terms of the trade-off of network complexity—measured as the number of nonzero parameters—and error, defined as the average per-character KL divergence of the LSTMs from the target distribution (the pFSAs from which the languages were sampled). For a target language with distribution $Q_G$ and a model $P_\theta$, this divergence is

$$\mathrm{D}[Q_G \| P_\theta] = \mathrm{H}[Q_G, P_\theta] - \mathrm{H}[Q_G]. \quad (4)$$

Thus, using the KL divergence here instead of cross entropy allows us to control for the entropy $\mathrm{H}[Q_G]$ of the target language $G$. We measure an approximate per-character KL divergence using the $N$ sample strings in the evaluation set for language $G$:

$$\hat{\mathrm{D}}[Q_G \| P_\theta] = \frac{1}{N} \sum_{i=1}^{N} \frac{1}{T_i} \sum_{t=1}^{T_i} \ln \frac{Q_G(x_i^t \mid x_i^{<t})}{P_\theta(x_i^t \mid x_i^{<t})}, \quad (5)$$

where $T_i$ is the length of the $i$'th sample string in the evaluation set, $x_i^t$ is the $t$'th character of the $i$'th sample string, and $x_i^{<t}$ is the prefix of characters up to index $t$ in the $n$'th sample string.

We use **area under the curve** (AUC) as a metric to evaluate complexity. This is defined as the area under the convex hull of (complexity, KL approximation) points. A small area under the curve indicates that it possible to achieve low error with a simple network; a large area under the curve means that error is high even with a more complex network. We determine if there is a significant difference between languages by performing a permutation test. This is done by randomly sampling points from the union of two languages and comparing the AUC statistic in the permuted samples.

### 4.1.2 Results

Figure 2 shows the trade-off of number of parameters and KL divergence for the three sets of languages. The areas under the curve are shaded, and the points which closely mimic the target distribution are shown in color. For a given level of complexity, the areas under the curve for the regular languages exceed both the piecewise testable and strictly local languages. The specific AUC numbers are shown in Table 3. In every case the

| Set | SL variant | SP variant | Regular variant |
|------|-----------|-----------|-----------------|
| $*aac$ | No string $aac$ | No sequence $a \ldots a \ldots c$ | No $c$ if the number of $a$ = 2 (mod 3) |
| $*ab, *ba$ | No string $ab$ / $ba$ | No sequence $a \ldots b$ / $b \ldots a$ | No $b$ ($a$) after an odd number of $a$ ($b$) |
| $*ac$ | No string $ac$ | No sequence $a \ldots c$ | No $c$ after an odd number of $a$ |

Table 2: Language sets composing the subregular battery. Each language set consists of three languages matched for FSA complexity: a strictly local language (SL), a strictly piecewise language (SP), and a regular language. All languages are subsets of $\{a, b, c\}^*$. The table gives a short description of each language. The last set is from Heinz and Idsardi (2013).

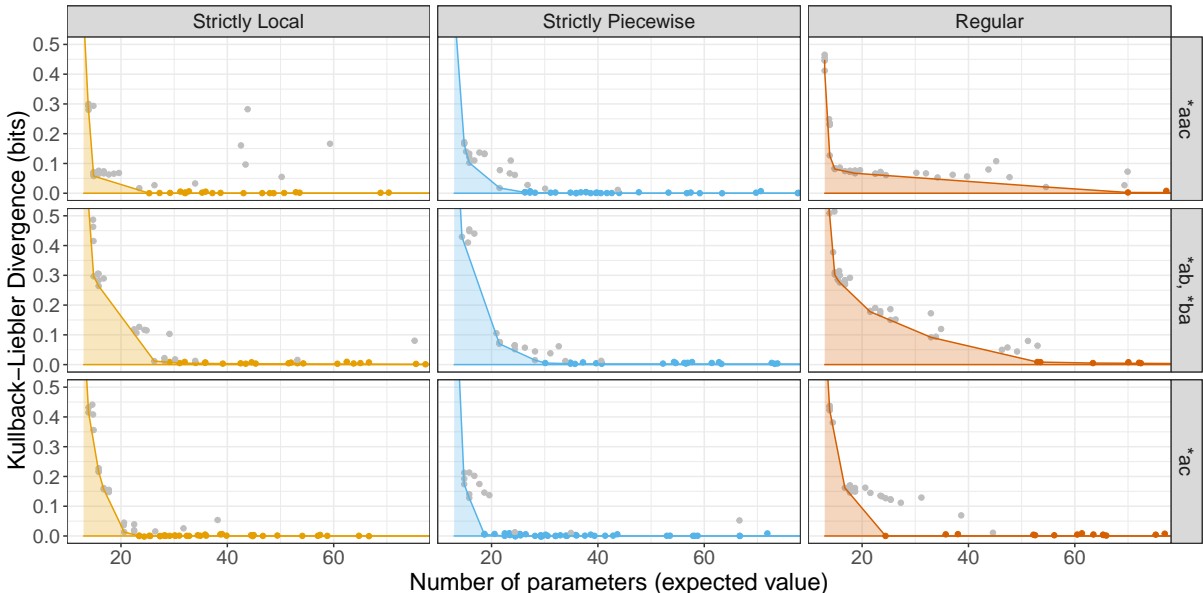

Figure 2: A trade-off of accuracy (KL divergence from the target language to the model) and complexity (number of nonzero parameters) of LSTMs trained on subregular and regular languages matched for description length in an FSA representation. Points within 1/100 of a bit or less from the true distribution are shown in color. The shaded area is the area under the curve (AUC) used to compare the languages in our statistical analysis in Table 3. The subregular languages show consistently lower AUC than the matched regular languages.

differences between both subregular languages and regular languages are statistically significant and in the expected direction as determined by a permutation test.

We augment this test with an additional post-hoc analysis of the most accurately performing networks (which we define as having a KL divergence of less than .01 bits) to determine if the size of the minimal LSTMs are also significantly different. This analysis is also done with a permutation test. These networks (Table 4) attain a lower complexity score for the subregular languages, a significant difference in all cases except *ac.

### 4.1.3 Discusssion

This experiment's results demonstrate that, at least in the languages chosen, strictly piecewise and strictly local languages better favor a tradeoff be-

|  | $*ac$ | $*aac$ | $*ab, *ba$ |
|------|-------|--------|-----------|
| Reg.−SL | .354** | 2.70** | 2.688** |
| Reg.−SP | .620** | 1.727** | 2.260** |
| SP−SL | −.266 | .970** | .428 |

Table 3: Differences in AUC between languages within the subregular battery, as shown in Figure 2. The first row shows the AUC difference for regular languages minus SL languages, etc. Comparisons marked ** are significant at $p < 0.01$ in a permutation test. The AUC differences between regular and subregular languages are always significant. Comparing SP and SL languages, the SP language has a significantly higher AUC for the *aac languages, and the difference is otherwise insignificant.

tween simplicity and complexity as demonstrated in our AUC analysis. We also see that they can be generated by functions implemented with simpler LSTMs in all but one case. In our post-hoc analysis we find that all but one subregular language is significantly simpler to accurately mimic in comparison with the regular language. These results also demonstrate that checking for the presence of two substrings is more complicated than one, and checking for a longer substring is more complicated than a shorter one. But if compared within a similar strictly regular pattern (here represented by controlling for generating FSA size) the preference is clearly for subregular languages.

|           | $*ac$    | $*aac$   | $*ab, *ba$ |
|-----------|----------|----------|------------|
| Reg.$-$SL | .981     | 44.7**   | 23.8**     |
| Reg.$-$SP | 5.74**   | 43.7**   | 22.9**     |
| SP$-$SL   | $-4.75*$ | .963     | .972       |

Table 4: Differences in complexity between languages within the subregular battery, as shown in Figure 2. The first row shows the complexity difference for regular languages minus SL languages, etc. Comparisons marked ** are significant at $p < 0.01$ in a permutation test. Complexity differences are significant between subregular and regular except for *ac*.

## 4.2 Standard Harmony and First-Last Harmony

In the second test we investigate whether a preference exists between two subregular languages. One language is MTSL, a class of subregular languages believed to be learnable in the subregular hypothesis. The other is a locally testable (LT) language, which is believed to *not* be human learnable. This test is inspired by previous experimental works (Lai, 2015; Avcu and Hestvik, 2020) which were investigations of standard vowel harmony (attested in human languages) and first-last harmony (not attested). These two languages are not controlled for FSA size, but are languages that serve as important evidence for the subregular hypothesis.

The difference between standard vowel harmony (SH) and first-last (FL) harmony languages is shown in Table 5. In an SH language we require languages to begin and end with an $a$ or $b$, and we forbid both $a$ and $b$ from occurring in the same string. In an FL harmony pattern, we require a word to begin and end with $a$ or $b$, and to begin and end with the same character, but allow any charac-

ters in-between. These examples are a simplistic representation of patterns that are attested in human language (standard harmony) versus not attested (first-last harmony).

In the experiment below, we apply a similar test and evaluation method as in the subregular battery to demonstrate that the functions which generate SH patterns (which are MTSL) are simpler than functions which generate FL patterns (which are LT).

| SH      | FL      |
|---------|---------|
| $bab$   | $bab$   |
| $cac$   | $cac$   |
| $babbab$| $bcabcb$|
| $cacacac$| $ccbabc$|
| $bbbbabb$| $bcacabb$|

Table 5: Examples of legal words in standard-harmony (SH) and first-last harmony (FL) patterns.

### 4.2.1 Methods

We use LSTMs with with the same architecture as in section 4.1. The only difference is that the objective for these networks is to predict the correct **continuation set**: the set of possible characters that can grammatically follow after a prefix, an objective introduced in Gers and Schmidhuber (2001) and used elsewhere (Suzgun et al., 2019). The network is trained to predict, for each possible character continuing a prefix, whether that character is licit or not—that is, whether that character is a member of the continuation set $C$. Given a continuation set $C$ and an LSTM equipped with a decoder that reads in a prefix $x$ of characters and produces a vector of real values $\in \mathbb{R}^{|C|}$, the loss for the network is

$$L_{\text{cs}}(p_\theta(x), C) = \frac{1}{|C|} \sum_{i \in |C|} -C_i \log(p_\theta(x)_i) - \quad (6)$$
$$(1 - C_i) \log(1 - p_\theta(x)_i)$$

This breaks with the prior work using the continuation set objective, as we use the binary cross-entropy loss instead of mean squared error.

Using this loss function is necessary to control for differences in entropy between the SH and FL languages when conducting comparisons. Above, we control for differences in entropy by using an approximate KL divergence as an accuracy metric and we control for length-related effects on LSTM

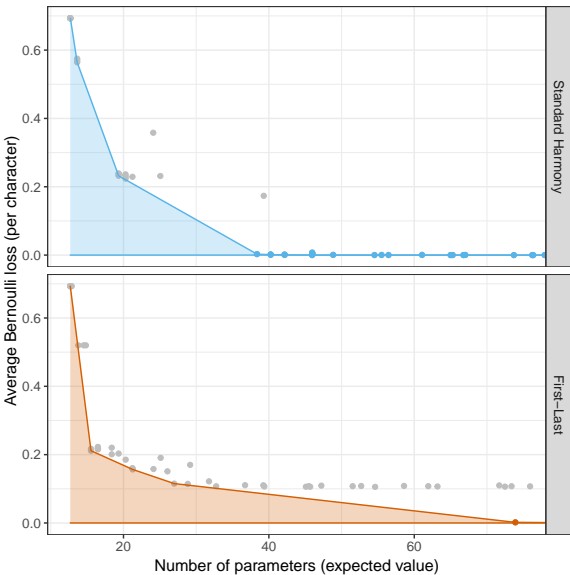

Figure 3: Graphs for LSTMs trained on first last harmony versus standard harmony, with description lengths on the x-axis, and test set performance on the y-axis (lower is better).

complexity by making the lengths of strings geometrically distributed. This is not possible when comparing the SH and FL languages, because they cannot be generated by pFSAs in a way that yields a geometric distribution on string length.

**Data generation**  To accommodate this new task, we sampled words uniformly given length. For SH data, words were generated such that they began and ended with the same character (either $b$ or $c$) with a geometrically sampled number of characters in-between ($p = .05$) consistent with the SH pattern (see table 5). For FL languages the same procedure was followed except that the center characters were consistent with the FL pattern.

**Network training**  Training was performed similarly to the subregular battery. This was for a maximum of 200 epochs, potentially terminating earlier if development set performance did not improve for 50 epochs.

**Evaluation**  At the end of training, (complexity, continuation-set loss) pairs were calculated from the best network's performance on the heldout test set. As in Section 4.1 we use AUC as the performance metric, the only difference being the points are now (complexity, continuation-set loss) instead of (complexity, KL approximation).

### 4.2.2  Results

Our results are shown in Figure 3. Again, we see a similar pattern as in the subregular battery, but this time showing the human-learnable subregular language is preferred over the unlearnable one. Standard harmony performance deterioriates at the 40 parameter mark, whereas first-last harmony deteriorates at the 75 parameter mark. The areas approximated by the frontier points (shaded) remain significantly different under a permutation test, with $p = .009$.

### 4.2.3  Discussion

These results continue the pattern seen in the subregular battery, but demonstrate not just a subregular preference, but a human-learnable subregular preference. While prior studies have shown a preference and higher performance for SH over FL with sequence to sequence networks, this experiment demonstrates that beyond a preference, these functions are more easily represented in simpler LSTMs.

### 4.3  Generalization

Above, we found that subregular languages, and particularly human-learnable ones, were represented by simpler LSTMs. The question remains: does this simplicity constraint affect generalization?

Here, we test whether a pressure for sparsity in LSTMs yields a bias towards learning a subregular language when the network is trained on data that is consistent with both a subregular and regular language, an approach suggested by Rawski et al. (2017). More precisely, the test is as follows. Take two formal languages $G_{\text{sub}}$ and $G_{\text{reg}}$, one subregular and one regular. Now, using their intersection $G_{\text{sub}} \cap G_{\text{reg}}$ as training data, what does the induced grammar correspond to, $G_{\text{sub}}$ or $G_{\text{reg}}$?

### 4.3.1  Methods

As our pair of langauges, we use languages from the subregular battery in Section 4.1. We take as $G_{\text{sub}}$ the language $*ac$ and as $G_{\text{reg}}$ the regular language which forbids $b$ after an odd number of $a$s. This language is the test as described in Rawski et al. (2017).

**Data generation**  For each network, we generate data from $G_{\text{sub}} \cap G_{\text{reg}}$, dividing these into a training set and a development set with approximately 1600 words in the training set and 200 words in

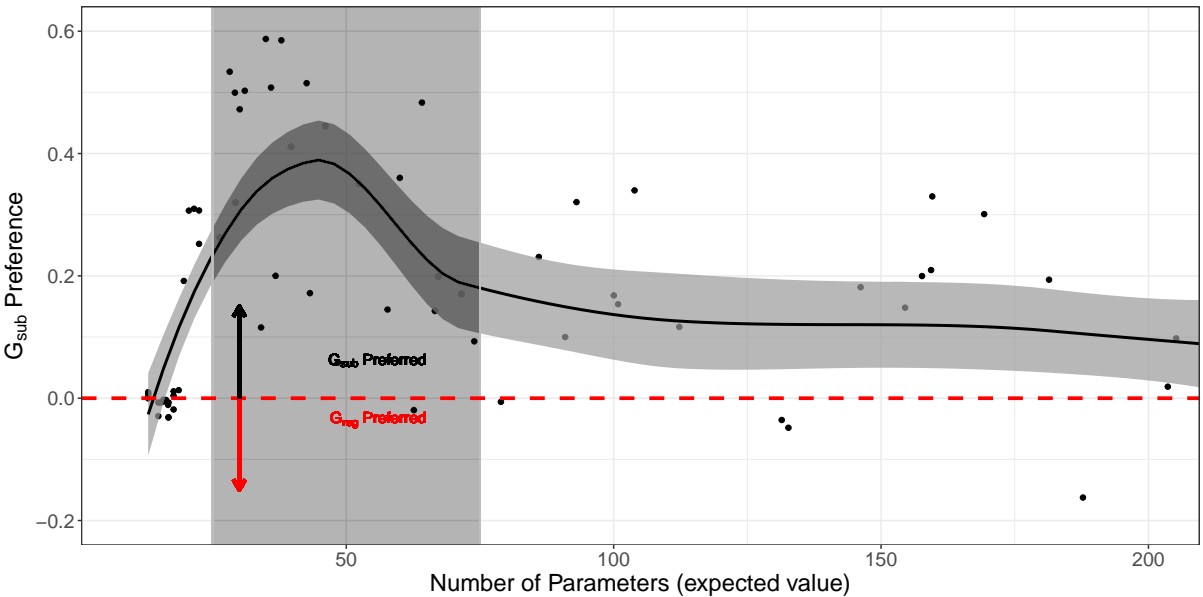

Figure 4: Model preference for subregular language $G_{\text{sub}}$ over regular language $G_{\text{reg}}$, where preference is defined according to Equation 7, as a function of the number of model parameters. The preference for $G_{\text{sub}}$ peaks in the shaded region occupied by simpler networks.

the testing set. We generate two additional testing datasets for $G_{\text{sub}}$ and $G_{\text{reg}}$ individually, checking the sets such that no words in the testing set appeared in the training or development data of 200 words each. This matches the size of the dataset to that of Section 4.1.

**Network training**   With the data generated, our LSTMs are trained according to the per character cross entropy loss, regularized as in Equation 3, with a procedure identical to that used in Section 2.

**Evaluation**   After training, performance on $G_{\text{sub}}$ and $G_{\text{reg}}$ is evaluated using the model which performed best on the development set over the course of training. As in the subregular analysis, we calculate the approximate KL divergence by subtracting the true per-character entropy as determined by the generating pFSA from the cross-entropy as in Equation 5.

Our analysis, unlike the previous ones, is qualitative. We will investigate the subregular bias, operationalized as the difference:

$$\text{Pref}_{G_{\text{sub}}} := \hat{\text{D}}[Q_{G_{\text{reg}}}\|P_\theta] - \hat{\text{D}}[Q_{G_{\text{sub}}}\|P_\theta], \quad (7)$$

where the approximate divergences $\hat{\text{D}}$ are calculated over the evaluation sets for the two languages $G_{\text{sub}}$ and $G_{\text{reg}}$, as in Eq. 5. The value $\text{Pref}_{G_{\text{sub}}}$ is positive when the model's distribution over strings more closely resembles the one defined by the pFSA for language $G_{\text{sub}}$.

### 4.3.2   Results

The results of our generalization experiment show a clear preference for the subregular language regardless of complexity (Figure 4). However, unlike prior results, we also see a clear increase in preferences as network complexity is constrained. This difference is especially pronounced in the highlighted region. In the lowest complexity range, there is a collapse in performance where the network can do nothing but predict a uniform distribution on all characters at all times.

Examining the KL divergences for the individual languages, as in Figure 5, we find that this result is driven by high accuracy in modeling the subregular language $G_{\text{sub}}$ using a small number of parameters. Accuracy in modeling the regular language $G_{\text{reg}}$ is mostly poor, while accuracy modeling subregular language $G_{\text{sub}}$ is highest when the expected number of parameters is between 25 and 75.

### 4.3.3   Discussion

While our prior results in the complexity evaluations have shown that subregular languages are generated with simpler networks, these results demonstrate a more clear inductive bias. LSTMs with constrained complexity generalize better to the subregular language than to the regular language. This indicates that, not only is the subregular function simpler, but that simpler networks prefer subregular languages.

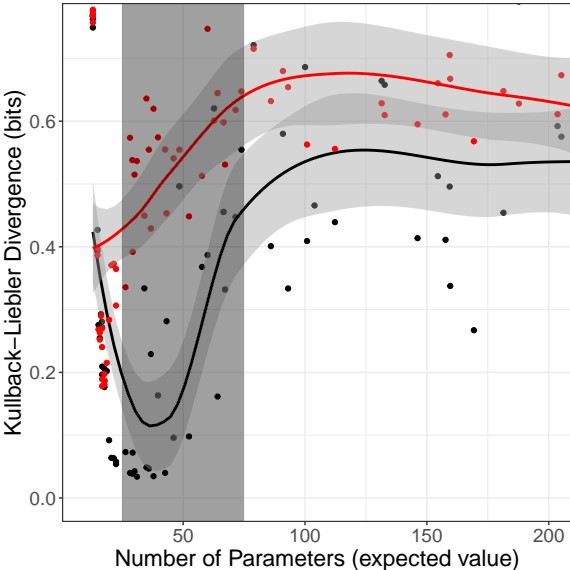

Figure 5: KL divergences from the target languages to the model distributions as a function of the number of model parameters. Performance on subregular languages (black) dips in the shaded region whereas performance on regular languages is relatively poor (red).

But we also see that this inductive bias exists even in low regularization regimes. This result is consistent with previous findings, and may be related to other phenomena like previously observed locality biases (Ravfogel et al., 2019) which are a basic constraint in all RNN functioning (Battaglia et al., 2018).

## 5 Conclusion

In our complexity measurement experiments we observe that subregular languages are represented by consistently simpler networks than similar regular languages, and that human-learnable subregular patterns are represented by simpler networks than unlearnable ones. This result is surprising, given that the use of a description length criterion to distinguish these languages had been deemed unlikely (Heinz and Idsardi, 2013).

Furthermore, we found that in the generalization experiment, data consistent with a subregular and regular language leads to a preferential generalization towards learning the subregular distribution, and that this preference increases as the network becomes simpler. This supports the idea that a change in computational representation systems may favor subregular over regular languages, with the caveat that comparing subregular languages with radically different complexities in FSA representation may

mean that this generalization pattern does not hold.

Why do LSTMs favor subregular languages? We argue that our results may explain this preference via the lottery ticket hypothesis (Frankle and Carbin, 2019). This hypothesis states that neural networks learn by tuning "winning tickets"—which are subnetworks that perform well on the task at initialization—which explains why larger neural networks perform well (they "buy more tickets"). The existence of smaller networks (as we define them) for subregular languages means that any neural network will contain more tickets for subregular languages at initialization, and thus have an inductive bias toward such languages. If this line of reasoning is true, $L_0$ regularization does not introduce anything qualitatively new in LSTM training, but shifts the probability mass to favor tickets with smaller subnetworks.

Perhaps more controversially, we also believe these results may be of interest to human cognition. While the human brain is not an LSTM, our results indicate that a subregular bias can be downstream of a bias for simplicity within an appropriate computational formalism, showing that rich (and often puzzling) constraints can be downstream from simple rules. A bias like this can be construed as a universal grammar—a bound on learnable formal languages (Nowak et al., 2002).

More needs to be done, of course. This work is far from a proof of the simplicity of subregular languages in LSTMs. Likewise, more subregular languages ought to be investigated. We may find, for example, that the true inductive bias for LSTMs is not exactly subregular, or that only certain aspects of subregularity are simple to implement. But we find the current results exciting and intriguing, for its relationship to network complexity and function implementation, its potential to explain LSTM inductive biases, and the demonstration of subregular biases resulting from computational complexity constraints.

## 6 Limitations

The largest limitation of this work is that it is experimental, and not a proof. We try to show several demonstrations of the simplicity of subregularity, and believe further work should be done, but are aware of the limitations of this kind of work in addressing matters of mathematical curiosity.

We also understand that, because this work addresses these issues in an experimental way, there

is no certainty that our object of study (subregular languages and their patterns) are what LSTMs are truly biased towards.

Our work uses LSTMs, rather than the Transformer architectures which underlie recently influential large language models (Brown et al., 2020). Although LSTMs are not currently the most popular architecture, recurrent models are recently seeing a revival (Peng et al., 2023) due to their inherent advantages such as being able to model indefinitely long sequences. In contrast to LSTMs, Transformers are known to be highly constrained in their strict formal expressivity (Hahn, 2020); however, their formal inductive biases in practice are not yet thoroughly explored.

## Ethical considerations

We foresee no ethical issues arising from this work.

## Acknowledgements

This work was supported by NSF grant #1947307 to R.F. We thank Connor Mayer, Jon Rawski, and Aniello de Santo for helpful discussion.

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

## A $L_0$ regularization as maximum a posteriori (MAP) estimation

We show that $L_0$ regularization is equivalent to MAP estimation with an exponential prior on the number of non-zero parameters in the model. Consider the standard MAP estimation problem:

$$\hat{\theta}_{\text{MAP}} = \arg\max_{\theta} f(x|\theta)q(\theta), \qquad (8)$$

where $f$ is a distribution conditional on parameters $\theta$ and $q$ is a prior distribution on $\theta$. Now consider the expected value of non-zero parameters $\theta'$. In Equation 3, $\theta'$ is the regularization term, as shown below:

$$\sum_{j=1}^{|\theta|} Q(z_j > 0 \mid \pi_j) \qquad (9)$$

$$= \sum_{j=1}^{|\theta|} \int q(z_j \mid \pi_j) \cdot 1\,(z_j > 0)\,\mathrm{d}z_j \quad (10)$$

$$= \sum_{j=1}^{|\theta|} \int q(z \mid \pi) \cdot 1\,(z_j > 0)\,\mathrm{d}z \qquad (11)$$

$$= \int q(z \mid \pi) \sum_{j=1}^{|\theta|} 1\,(z_j > 0)\,\mathrm{d}z \qquad (12)$$

$$= \mathop{\mathbb{E}}_{z \sim Q(\cdot|\pi)} \left[ \sum_{j=1}^{|\theta|} 1\,(z_j > 0) \right] \qquad (13)$$

$$= \theta', \qquad (14)$$

with probability density function $q(\cdot \mid \pi)$ corresponding to cumulative distribution function $Q(\cdot \mid \pi)$, and indicator function $1(\cdot)$. If we consider an exponential prior on $\theta'$ we would have:

$$q(\theta') = \lambda e^{-\lambda\theta'}. \qquad (15)$$

Then by Eq. 15 and the monotonicity of the $\log$ function, the MAP estimation problem is equivalent to finding

$$\hat{\theta}_{\text{MAP}} = \arg\min_{\theta} -\log f(x|\theta) + \lambda\theta' - \log\lambda, \quad (16)$$

which is equal to our objective in Eq. 3 up to an additive constant $-\log\lambda$ that does not depend on the parameters to be optimized.