# OpenReview forum: "Simpler neural networks prefer subregular languages"
_EMNLP/2023/Conference — EMNLP 2023 Findings_

### Official Review · Reviewer_eZgg · 2023-08-04

**Soundness:** 4

**Excitement:**

4: Strong: This paper deepens the understanding of some phenomenon or lowers the barriers to an existing research direction.

**Paper Topic And Main Contributions:**

The paper shows that learning a sparse LSTM in a subregular language is more successful than in a regular language. In a sense, this shows a bias towards the simplicity of the LSTM _trained by the gradient-based method_. The paper compares the performance of LSTM 1) in subregular versus regular languages, 2) between two subregular languages, 3) as well as its generalizing ability in subregular versus regular languages. The analysis was carried out empirically, but at the same time thoroughly.

**Questions For The Authors:**

A) lines 134--137: This sentence is confusing. Is Q supported on {0, 1}? Does sampling z from Q produce either 0 or 1? Or can z take any value in the interval [0, 1] ?

**Reasons To Accept:**

- The work explores the bias towards simplicity, which is the basis for the successful training of modern learning algorithms. Unlike other works in this area, the focus here is on the nature of human language and its relationship to subregular languages.


**Reasons To Reject:**

- Personally, I lacked more pedagogical coverage of subregular languages in Section 2. The work seems to imply that the reader is already familiar with this area, but this is not entirely true for many potential readers of the article.

**Reproducibility:**

4: Could mostly reproduce the results, but there may be some variation because of sample variance or minor variations in their interpretation of the protocol or method.

**Reviewer Confidence:**

3: Pretty sure, but there's a chance I missed something. Although I have a good feel for this area in general, I did not carefully check the paper's details, e.g., the math, experimental design, or novelty.

---

> ### Author Rebuttal · Authors · 2023-08-29
>
> Thank you for your review and your time. We will address what we can in the paper. As space is limited, we may make a point of addressing readers to the respective sources after giving a brief introduction to them.
>
> >*Personally, I lacked more pedagogical coverage of subregular languages in Section 2. The work seems to imply that the reader is already familiar with this area, but this is not entirely true for many potential readers of the article.*
>
> This is a fair point. We will devote some space in the final paper to introducing the notion of subregularity and the subregular classes under investigation for the sake of future readers.
>
> >*A) lines 134--137: This sentence is confusing. Is Q supported on {0, 1}? Does sampling z from Q produce either 0 or 1? Or can z take any value in the interval [0, 1] ?*
>
> Q is supported on the interval $[0, 1]$, however it is defined such that the distribution has a non-zero probability of sampling the values of 0 and 1 in particular. We did not want to spend too much space on this topic, as we are just reintroducing the method from Louizos (2017). The distribution they (and we) use is defined in the following way.
>
> First, take a binary concrete random variable S parameterized by location $log(\alpha)$ and temperature $\beta$. This variable has support on the open interval $(0, 1)$, and can also be fit using the reparameterization trick. Then, “stretch” this distribution $\bar{s} = s(\zeta - \gamma) + \gamma$. Finally, clip this distribution  so it is again within the interval specified $min(1, max(0, \bar{s}))$. This procedure results in obtaining samples from the closed interval $[0, 1]$ with non-zero probabilities for obtaining 0 or 1. This is explained in more detail in section 2.2 of Louizos (2017).
>
> We will make the supported interval clearer in the text, although we want to avoid devoting too much space to explaining the method, as it is not our own contribution.

---

### Official Review · Reviewer_ABqy · 2023-08-06

**Typos Grammar Style And Presentation Improvements:** NA
**Soundness:** 3

**Excitement:**

4: Strong: This paper deepens the understanding of some phenomenon or lowers the barriers to an existing research direction.

**Missing References:**

NA

**Paper Topic And Main Contributions:**

The authors test whether LSTM's have a  bias for -- thus can better learn -- sub-regular languages than regular ones. Within the answer set, they check for different types of sub-regularity, namely minimal description versus local and strict descriptions. They also check for sub-regular languages that were better aligned to human taste. In each occasions the answer is yes. LSTM's seem to prefer human-related sub-regular language to non human-related ones, to regular ones.

**Questions For The Authors:**

1- I am not sure if I understand the connection between networks that are biased towards sub-regular languages  and sparsity, e.g. as you claim on lines 053 and 054 on page 1. Why in order to study simplicity we need to work with sparsity? Can you please clarify?

-- a little while down, you do tell us why, that you would like ot use the L0 norm, and this is a reason. But can you now tell us why
   1.1. why would you like to use L0 norm?
    1.2. why the choice of a norm is related to the simplicity of a neural network?

2- On page 1, line 092, can you please tell us what a minimum description language is?

3- After equation (3) on page 2, you discuss the different interpretation for L, when it is cross entropy Firstly, do other cost functions also have different interpretations? If not, why cross entropy is special? Secondly, please can you tell us where do these two interpretations come from and how do they relate to your main concept, namely MDL?

4- On page 4, line 187, you talk about  the importance of using a geometric distribution on word length when generating your data. But do not tell us why this is important and why a geometric distribution is used? Is it, for instance, related to Zipf's law etc? Why not use a Gaussian for instance, which is the more standard one?

4- On page 4, lines 262 - 264, you conclude  simpler LSTM's can generate  strictly piecewise and strictly local languages, but the table of results only shows that the AUC is better for these languages in comparison with regular languages. Can be more precise in your statement?

**Reasons To Accept:**

The idea is original and interesting, it would be great if other tests could be done and other people could regenerate the results so we were sure of the results, that LSTM's indeed learn the sub-regular human aligned languages better. The paper is well written and explains the ideas and results well.

**Reasons To Reject:**

Something that bothered me all along was the connection between the main idea, that LSTM's learn sub-regular better than regular, and sparsity. Why are these two related?

The other issue, for me, was the main objective function and its relationship to cross entropy and other cost functions. This has to be mapped out better, see my questions below.

**Reproducibility:**

4: Could mostly reproduce the results, but there may be some variation because of sample variance or minor variations in their interpretation of the protocol or method.

**Reviewer Confidence:**

3: Pretty sure, but there's a chance I missed something. Although I have a good feel for this area in general, I did not carefully check the paper's details, e.g., the math, experimental design, or novelty.

---

> ### Author Rebuttal · Authors · 2023-08-29
>
> Thank you for your review and your time. We address your questions below, and will add clarifications to the paper or to make the introductions of these topics better.
>
> >*1- I am not sure if I understand the connection between networks that are biased towards sub-regular languages and sparsity, e.g. as you claim on lines 053 and 054 on page 1. Why in order to study simplicity we need to work with sparsity? Can you please clarify? -- a little while down, you do tell us why, that you would like to use the L0 norm, and this is a reason. But can you now tell us why 1.1. why would you like to use the L0 norm? 1.2. why the choice of this norm is related to the simplicity of a neural network?*
>
> The L0 norm is desirable for two reasons. First, it results in networks that are smaller in the sense of having fewer parameters. This corresponds directly to a notion of “description length” for a network and so connects to MDL methods. Second, the L0 norm may reveal inductive biases of neural networks via the lottery ticket hypothesis. This hypothesis states that neural networks tune “winning tickets”–subnetworks that perform well on the task at initialization–which explains why larger neural networks perform well (they “buy more tickets”). While this is not the main thrust of this paper, the existence of smaller networks (as we define them) for subregular languages can explain already observed inductive biases: any neural network will contain more “tickets” for subregular languages at initialization. We will add text explaining these points to the conclusion.
>
>
> >*2- On page 1, line 092, can you please tell us what a minimum description language is?*
>
> The Minimum Description Length method is a way of rigorously employing the notion of simplicity in learning. It is a mathematically specified version of Occam’s razor: that the best hypothesis to account for a given dataset is the simplest one. This is done using a two part code, one part encoding the distribution and another part encoding the observations, which is a form of MDL which is flexible and applicable to diverse model classes (including neural networks, Grunwald (2000)). This is the source of our objective function, and the interpretation consistent with the MDL literature. We will add text on this to Section 2.
>
> >*3- After equation (3) on page 2, you discuss the different interpretation for L, when it is cross entropy Firstly, do other cost functions also have different interpretations? If not, why cross entropy is special? Secondly, please can you tell us where do these two interpretations come from and how do they relate to your main concept, namely MDL?*
>
> The cross entropy of a distribution P (for example, the true distribution in a training set) and Q (for example, a model distribution) can be interpreted as the cost of encoding data from distribution P under distribution Q, in terms of the number of bits required. For this reason, minimizing cross entropy is equivalent to finding a minimal-length description of P, and also equivalent to maximizing the likelihood of the data. In our case, we are adding a regularization term, which means we are searching for a minimal-length encoding of both the model and the data. We will add text explaining this point to Section 3.
>
> >*4- On page 4, line 187, you talk about the importance of using a geometric distribution on word length when generating your data. But do not tell us why this is important and why a geometric distribution is used? Is it, for instance, related to Zipf's law etc? Why not use a Gaussian for instance, which is the more standard one?*
>
> We will add text explaining the importance of the geometric distribution on lengths: “The geometric distribution on word length is important for two reasons: (1) it implies a constant stopping probability in all states, meaning that the distribution over string lengths does not depend on the structure of the strings in a language, and (2) it is memoryless, meaning that the network need not keep track of the number of characters observed so far to attain high accuracy. Maintaining a geometric distribution on string lengths is the simplest method for attaining the same distribution over string lengths across languages while sampling from pFSAs.”
>
> >*5 - On page 4, lines 262 - 264, you conclude simpler LSTM's can generate strictly piecewise and strictly local languages, but the table of results only shows that the AUC is better for these languages in comparison with regular languages. Can be more precise in your statement?*
> In accordance with this comment, we performed an additional statistical analysis on LSTMs which achieved high accuracy (within .01 bits of 0). The results are below:
>
>
> |   |  \*ac | \*aac| \*ab, \*ba |
> |---|---|---|---|
> | Reg - SL | .981 | 44.7\*\*|23.8\*\*|
> | Reg - SP| 5.74\*\*|43.7\*\*|22.9\*\*|
> | SP-SL| -4.75\*|.963|.972|
>
> We will add this data in and qualify our statements in accordance with this analysis, as results were significant and in the expected direction between all pairings except for Reg-SL on *ab, which was not statistically significant. We will also amend the language in those results to reflect instead the favorable accuracy/complexity tradeoff that the LSTMs showed for the subregular languages.

---

### Official Review · Reviewer_Vqqg · 2023-08-08

**Soundness:** 4

**Excitement:**

4: Strong: This paper deepens the understanding of some phenomenon or lowers the barriers to an existing research direction.

**Paper Topic And Main Contributions:**

This paper looks at sparsity patterns as a means of studying inductive biases in LSTMs for formal languages. In general, this is a very well-written paper that covers formal language theory and gives interesting insights into how subregular languages are represented by simpler networks. The paper motivates why LSTMs are still an interesting problem and why that can relate to more popular neural methods such as the lottery ticket hypothesis. Overall, I would be happy to see this paper at EMNLP as it gives some interesting insights into formal language theory, neural networks, and demonstrates some empirically insightful results. In addition, I think it will have longer lasting power than many of the recent papers in the field that maximize a metric without any significant insights.

**Questions For The Authors:**

- What happens as you scale the network architecture? Would a much larger network still exhibit the same behavior? Would the sparsity patterns still be seen with the L0 regularizer?
- Along similar lines, what would happen if you sampled more than 2,000 words from the pFSA? How does this change the analysis?

**Reasons To Accept:**

- interesting insights into formal language theory in relation to neural networks
- very well-written. Clear and concise for a difficult topic

**Reasons To Reject:**

- the LSTM model is quite small. "a 3-dimensional embedding layer, a 5 unit LSTM layer, and a 2-layer perceptron decoder with a hidden width of 5 units"


**Reproducibility:**

4: Could mostly reproduce the results, but there may be some variation because of sample variance or minor variations in their interpretation of the protocol or method.

**Reviewer Confidence:**

2: Willing to defend my evaluation, but it is fairly likely that I missed some details, didn't understand some central points, or can't be sure about the novelty of the work.

---

> ### Author Rebuttal · Authors · 2023-08-29
>
> Thank you for your review and your time. We try to address your questions below.
>
> >*What happens as you scale the network architecture? Would a much larger network still exhibit the same behavior? Would the sparsity patterns still be seen with the L0 regularizer? Along similar lines, what would happen if you sampled more than 2,000 words from the pFSA? How does this change the analysis?*
>
> Both of these are good questions. Unfortunately, the method we used (the Louizos (2017) method) is computationally costly. That, paired with having to run the training many times at different regularization penalties means we were forced to work with smaller networks and datasets. Presently, we are experimenting with faster means of approximating the L0 norm, however we have not been able to run the experiment at a larger size. At least on wider networks, any difference in results would be due to priors imposed by the gradient descent algorithm, as the objectives we used were based on an unnormalized number of parameters.

---

### Meta-Review · Area_Chair_8tFT · 2023-09-19

**Recommendation:** 4

**Metareview:**

The methodology, results, and presentation are sound. The relevance to linguistics and human cognition are slimmer.

---

### Decision · Program_Chairs · 2023-10-07

**Decision:**

Accept-Findings

**Comment:**

The methodology, results, and presentation are sound. The relevance to linguistics and human cognition are slimmer.